# 68Ga-PSMA-PET/CT-based radiosurgery and stereotactic body radiotherapy for oligometastatic prostate cancer

Goda Kalinauskaite[1,2]*, Carolin Senger[1,2], Anne Kluge[1,2], Christian Furth[3], Markus Kufeld[2], Ingeborg Tinhofer[1,4], Volker Budach[1,2], Marcus Beck[1,2], Alexandra Hochreiter[1]⦵, Arne Grün[1,2]⦵, Carmen Stromberger[1,2]⦵

1 Charité –Universitätsmedizin Berlin, corporate member of Freie Universität Berlin, Humboldt-Universität zu Berlin, and Berlin Institute of Health, Department of Radiation Oncology and Radiotherapy, Berlin, Germany, 2 Charité CyberKnife Center, Departments of Radiation Oncology and Neurosurgery, Charité - Universitätsmedizin Berlin, Berlin, Germany, 3 Charité-Universitätsmedizin Berlin, corporate member of Freie Universität Berlin, Humboldt-Universität zu Berlin, and Berlin Institute of Health, Department of Nuclear Medicine, Berlin, Germany, 4 The Translational Radiooncology and Radiobiology Research Laboratory, Charité - Universitätsmedizin Berlin, Berlin, Germany

⦵ These authors contributed equally to this work.
* goda.kalinauskaite@charite.de

**Data Availability Statement:** All relevant data are within the manuscript and its Supporting Information files.

## Abstract

### Background

Androgen deprivation therapy (ADT) remains the standard therapy for patients with oligometastatic prostate cancer (OMPC). Prostate-specific membrane antigen positron emission tomography/computed tomography (PSMA-PET/CT)-based stereotactic body radiotherapy (SBRT) is emerging as an alternative option to postpone starting ADT and its associated side effects including the development of drug resistance. The aim of this study was to determine progression free-survival (PFS) and treatment failure free-survival (TFFS) after PSMA-PET/CT-based SBRT in OMPC patients. The efficacy and safety of single fraction radiosurgery (SFRS) and ADT delay were investigated.

### Methods

Patients with ≤5 metastases from OMPC, with/without ADT treated with PSMA-PET/CT-based SBRT were retrospectively analyzed. PFS and TFFS were primary endpoints. Secondary endpoints were local control (LC), overall survival (OS) and ADT-free survival (ADTFS).

### Results

Fifty patients with a total of 75 metastases detected by PSMA-PET/CT were analyzed. At the time of SBRT, 70% of patients were castration-sensitive. Overall, 80% of metastases were treated with SFRS (median dose 20 Gy, range: 16–25). After median follow-up of 34 months (range: 5–70) median PFS and TFFS were 12 months (range: 2–63) and 14 months (range: 2–70), respectively. Thirty-two (64%) patients had repeat oligometastatic disease. Twenty-four (48%) patients with progression underwent second SBRT course. Two-year LC after SFRS was 96%. Grade 1 and 2 toxicity occurred in 3 (6%) and 1 (2%) patients,

**Funding:** Award was received by GK. Funder: Berliner Krebsgesellschaft das Ernst von Leyden-Stipendium URL: https://www.berliner-krebsgesellschaft.de/krebsforschung/ernst-von-leyden-stipendium/ The funders had no role in study design, data collection and analysis, decision to publish, or preparation of the manuscript.

**Competing interests:** The authors have declared that no competing interests exist.

respectively. ADTFS and OS rates at 2-years were 60.5% and 100%, respectively. In multivariate analysis, TFFS significantly improved in patients with time to first metastasis (TTM) >36 months (p = 0.01) and PSA before SBRT ≤1 ng/ml (p = 0.03).

## Conclusion

For patients with OMPC, SBRT might be used as an alternative to ADT. This way, the start/escalation of palliative ADT and its side effects can be deferred. Metastases treated with PSMA-PET/CT-based SFRS reached excellent LC with minimal toxicity. Low PSA levels and longer TTM predict elongated TFFS.

## Introduction

For stage IV prostate cancer (PCA) palliative systemic therapy with androgen deprivation and/or chemotherapy with docetaxel remains the standard of care [1]. However, some patients with a limited number of metastases have a less aggressive disease course and might be treated with metastasis directed therapy (MDT) for all tumor sites as an alternative to systemic treatment [2]. These patients represent a condition known as oligometastatic disease, which is defined as an intermediate state between localized cancer and widespread metastases [3]. In the context of oligometastatic prostate cancer (OMPC), the desired effect of MDT is to postpone the start or escalation of androgen deprivation therapy (ADT) or in some cases even to achieve long lasting remission [4]. As a result, delayed onset of ADT-associated side effects and the inevitable emergence of therapy resistant PCA can be assumed.

The advent of positron emission tomography (PET) with different tracers has improved the diagnosis of patients with OMPC by detecting early recurrence. The prostate-specific membrane antigen (PSMA) is a membrane-specific type II glycoprotein that is overexpressed in more than 80% of PCA cells and is therefore an ideal target for diagnostic imaging [5, 6]. Recently Gallium-68-labelled PSMA PET computed tomography (PSMA-PET/CT) was found to be superior in localizing actively metabolizing tumor in patients with primary diagnosis or recurrence of PCA compared to conventional imaging modalities and choline-based PET/CT [7–11]. The detection rates for PSMA-PET/CT reported in the literature vary from 46% to 97% depending on the levels of prostate-specific antigen (PSA) [12–15]. Some authors observed detection rates of >50% in patients with PSA <0.5 ng/mL [16, 17]. Such a high sensitivity allows identification of very early recurrences with lesions <5 mm in size [10].

One-year local control (LC) rates reported after fractionated stereotactic body radiotherapy (fSBRT) for patients with oligometastatic prostate cancer vary from 93–100%. Besides, no grade ≥3 adverse events have been observed [18–20]. In this regard, single fraction radiosurgery (SFRS) is particularly attractive, since LC rates seem to be equally effective but treatment is delivered in a single session [21].

The primary aim of this study was to assess progression-free survival (PFS) and treatment failure free-survival (TFFS) after PSMA-PET/CT-based SFRS or fSBRT in patients with OMPC with up to five metastases. Further endpoints included safety and efficacy of SFRS, overall survival (OS) and possible delay of ADT initiation.

## Materials and methods

### Study population

In this retrospective analysis men with de-novo oligometastatic PCA (synchronous oligometastatic disease or metachronous oligorecurrence or metachronous oligoprogression) who

received curative 68Ga-PSMA-PET/CT-based SBRT for all metastases were included [22]. No more than 5 metastases in ≤3 organs were allowed. The first metastasis was diagnosed after median time of 37 months (1–199) from the initial diagnosis of PCA. All men had curative therapy for prostate cancer. Both castration sensitive and castration resistant patients were eligible for this study. The patients who started ADT and SBRT at the same time and patients with previous SBRT were excluded.

This single center study was approved by the institutional medical ethics committee of the Charité-Universitätsmedizin Berlin (EA1/214/16).

### Radiotherapy

SBRT/SFRS was performed using mainly the CyberKnife (CK) Robotic Radiosurgery System (Accurray®, USA) and dedicated stereotactic linear accelerator. CK Fiducial® Tracking (Accurray®, USA) was applied if indicated (e.g. lymph nodes expected to shift independently to the bone) with one gold fiducial (1.0 mm x 5.0 mm) being placed within/close to the target under CT guidance. Otherwise, patients were aligned to the spine using XsightSpine® Tracking (Accuracy®, USA) or ExacTrac-based spine alignment (BrainLab®, Germany). A thin-slice planning CT with 1.0–2.0 mm slices in supine position was obtained. PSMA-PET/CT images were co-registered for contouring. The gross tumor volume was contoured on all axial CT slices. The clinical target volume corresponded to the gross tumor volume. The planning target volume was created by adding a 2–5 mm margin around the clinical target volume. A SFRS/fSBRT dose was prescribed to the 70–80% isodose surrounding the planning target volume (Fig 1).

The fractionation regiments were selected taking into account the location of the lesion. If the irradiated metastasis was in the immediate vicinity of the organs at risk and therefore dose restrictions could not be met, fSBRT was indicated. Otherwise, SFRS was preferred over fSBRT for patient comfort, economic and logistic advantages.

### Follow-up

Follow-up was obtained every 3 months after SBRT within the first two years and half-yearly thereafter. Adverse events were scored using the National Cancer Institute Common Toxicity Criteria version 4. Additionally, patients attended routine follow-up visits at their urologist.

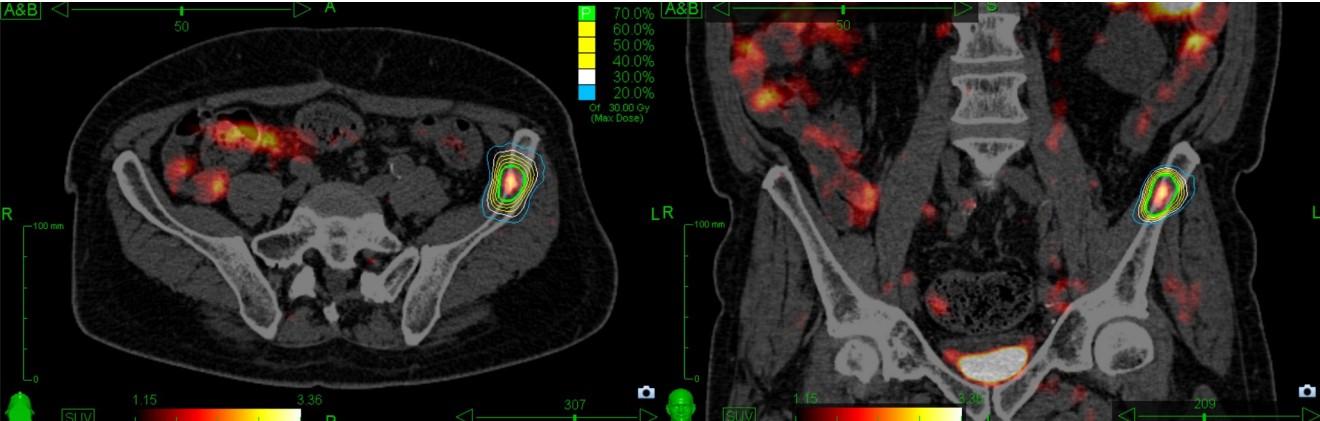

**Fig 1. PSMA-PET/CT based radiotherapy treatment plan of CyberKnife treatment system for bone metastasis located in the left ilium.**

### Endpoints

Endpoints of the study were PFS, TFFS, local control (LC), ADT-free survival (ADTFS), ADT-escalation-free survival (ADTEFS) and OS calculated from the start of SBRT. PFS was defined as freedom from biochemical failure, in-field progression, distant metastases or death. For TFFS new tumor-directed therapy (e.g. repeated SBRT, start of ADT, escalation of an ongoing ADT, surgery, chemotherapy) or death were determined as events. For LC, the in-field progression was counted as an event and was defined as an increase of metastasis volume or local regrowth within the PTV. LC was assessed using conventional (CT or MRT) or functional (PSMA-PET/CT) imaging. ADTFS was the interval until onset of ADT or death, whereas ADTEFS was defined as time to ADT-escalation or death for patients with ongoing ADT. For OS death of any cause was determined as an event.

### Statistical analysis

Survival analysis was conducted using the Kaplan–Meier method. The Cox proportional hazard model was used in univariate and multivariate analyses to calculate hazard ratios (HR) with 95% confidence intervals (95% CI). Covariates with a p-value $\leq 0.1$ in univariate analysis were included in the multivariate analysis. The Chi-square test was performed to compare variables. A p-value of $<0.05$ was considered to be statistically significant. Data processing and statistical analysis were conducted using FileMaker Pro 15 Advanced, Excel 2010 and IBM SPSS Statistics 24 (SPSS Inc., Chicago, IL, USA).

## Results

Between January 2012 and December 2016, 50 patients with OMPC and 75 oligometastases detected by PSMA-PET/CT were treated with SBRT to all tracer-avid metastatic lesions. Patients, metastases, and treatment characteristics are summarized in Table 1 and S1 Table. At the initial diagnosis of PCA, 41 patients (82%) were classified as high risk according to the D'Amico classification [23]. Three (6%) and 4 (8%) patients had low- and intermediate-risk PCA, respectively. In 2 (4%) patients the risk class was unknown. Fifteen patients (30%) were castration resistant. Median time from PCA diagnosis to the first metastasis (TTM) was 37 months (range: 1–199). Forty-eight (96%) patients had single organ involvement. The median number of metastases treated per patient was one (range: 1–5). SFRS with a median PTV-surrounding dose of 20 Gy (range: 16–25) was applied to 60 (80%) metastases, 13 (17.3%) received fSBRT with 24 Gy in 3 fractions (3 x 8 Gy) and 2 other schedules (2.7%) (S2 Table).

With a median follow-up of 34 months (range: 5–70), the 1-, 2-years PFS and TFFS were 54%, 22%, and 55.2%, 23.4%, respectively (Fig 2A and 2B). Median PFS and TFFS were 12 months (95% CI: 7.6–16.3) and 14 months (95% CI: 10–17.9), respectively. The TFFS significantly improved in patients with time to first metastasis >36 months (Fig 2C). Progression occurred in 49 patients (98%), with 32 patients (64%) having repeat oligometastatic disease with median two new metastases (range: 1–5). Forty-two (84%) patients underwent repeated PSMA-PET/CT due to a rising PSA. Treatment failure was observed in 46 patients (92%). Of these, 24 patients (48%) were treated with a second course of PSMA-PET/CT-based SBRT. The median time from the first to the second course of SBRT was 17 months (95% CI: 9.7–24.2). Fourteen patients (28%) started ADT, whereas in 6 patients (12%) ADT was escalated. The pattern of progression and new tumor-directed therapies is presented in Table 2. At the last follow-up, 31 (62%), 13 (26%), 3 (6%), and 1 (2%) patients had 2, 3, 4, and 5 courses of SBRT, respectively.

Local control was available for 73 lesions. The 1-, 2-year LC rates after SFRS and fSBRT were 98%, 96% and 100%, 100%, respectively (Fig 2D). There was no significant difference

**Table 1. Patients, tumor and treatment characteristics.**

| Characteristic | Value |
|---|---|
| **Age at PCA diagnosis, years** | |
| Median (range) | 62 (47–75) |
| **PSA at PCA diagnosis, ng/mL** | |
| Median (range) | 9.8 (0.54–159) |
| **PSA at SBRT, ng/mL** | |
| Median (range) | 1.9 (0.16–59.8) |
| **Gleason score, N (%)** | |
| ≤6 | 3 (6) |
| 7 | 28 (56) |
| ≥8 | 18 (36) |
| unknown | 1 (2) |
| **Primary tumor size (T), N (%)** | |
| c/pT1-T2b | 16 (32) |
| c/pT2c-T3 | 32 (64) |
| Tx | 2 (4) |
| **Regional lymph node involvement at PCA diagnosis, N (%)** | |
| c/pN0 | 36 (72) |
| c/pN1 | 11 (22) |
| Nx | 3 (6) |
| **PCA treatment, N (%)** | |
| RP | 15 (30) |
| RT | 4 (8) |
| RP and RT | 31 (62) |
| **ADT at the time of SBRT, N (%)** | |
| no | 35 (70) |
| yes | 15 (30) |
| **Time to metastases from diagnosis of PCA (months)** | |
| Median (range) | 37 (1–199) |
| **Number of metastases treated at first SBRT, N (%)** | |
| 1 | 35 (70) |
| 2 | 9 (18) |
| 3 | 3 (6) |
| 4 | 2 (4) |
| 5 | 1 (2) |
| **Primary site of metastases, N (%)** | |
| Lymph node | 24 (48) |
| Pelvic | 15 (62.5) |
| Extra-pelvic | 8 (33.3) |
| Both | 1 (4.2) |
| Bone | 23 (46) |
| Bone and lymph node | 2 (4) |
| Lung | 1 (2) |
| **Maximal SUV of PSMA-PET/CT** | |
| Median (range) | 6 (2.6–42) |
| **Fractionation schedules, N (%)** | |
| SFRS | 60 (80) |
| 3 fractions | 13 (17.3) |

(*Continued*)

**Table 1.** (Continued)

| Characteristic | Value |
|---|---|
| other | 2 (2.7) |
| **Median dose (Gy) for SFRS (range)** | 20 (16–25) |
| **Median dose (Gy) for fSBRT(range)** | 24 (19.2–28.8) |

Abbreviations: ADT = androgen deprivation therapy; fSBRT = fractionated stereotactic body radiotherapy; PCA = prostate cancer; PSA = prostate-specific antigen; PSMA-PET/CT = prostate-specific membrane antigen positron emission tomography/computed tomography; RP = radical prostatectomy; RT = radiotherapy; SBRT = stereotactic body radiotherapy; SFRS = single fraction radiosurgery; SUV = standardized uptake value.

observed for LC in SFRS and fSBRT groups (p = 0.55). Two (2.7%) bone metastases relapsed after SFRS with 20 Gy and 21 Gy, respectively. One was repeatedly treated with fSBRT.

At the last follow-up, 42.9% (15/35) of primarily ADT-naïve patients started treatment with ADT. The 1- and 2-year rates for ADTFS were 76.4% and 60.5%, respectively. Median ADTFS was not reached. ADT escalation was performed in 73.3% (11/15) of patients, with 1- and 2-year ADTEFS rates being 58.2% and 33.9%, respectively. The median ADTEFS was 27 months (95% CI: 8.8–45.1). Four patients were dead at the time of analysis. 1-, 2- and 5-years OS rates were 100% and 100% and 80.3%. Median OS was not reached (Fig 2E). There was a trend towards better OS in patients treated with the second course of SBRT compared to patients receiving other therapy (Fig 2F).

Results of univariate and multivariate analysis of clinical prognostic factors affecting PFS and TFFS are summarized in Table 3. In multivariate testing, a TTM >36 months (p = 0.01) and PSA ≤1 ng/ml before SBRT predicted significantly longer TFFS (p = 0.03). In addition, a longer PFS in univariate analysis was observed (p = 0.01) in patients with a TTM >36 months. Multivariate analysis for PFS was not conducted because only one covariate had a p-value ≤0.1.

Acute grade 1 toxicity was observed in three (6%) patients: 1 fatigue, 1 pain within the irradiated region, and 1 subacute pneumonitis. Only 1 (2%) grade 2 fatigue was observed. No grade 3 or higher acute or any late toxicity occurred. No significant differences in terms of toxicities between SFRS and fSBRT were observed (p = .58).

## Discussion

This study complements the existing literature on metastases-directed therapy (MDT) for patients suffering from OMPC in several ways. First, we analyzed a large number of metastases treated with PSMA-PET/CT based SFRS. Second, we reported outcomes after repeated use of SBRT with the intention to defer the start or escalation of palliative ADT.

To the best of our knowledge, there are only two randomized studies that examined MDT in comparison to observation for OMPC patients. In the STOMP Phase 2 trial, either SBRT with 10 Gy in 3 fractions or surgery was used after staging with choline PET/CT [2]. Lately announced 5-year follow-up results showed significantly lower rates of ADT onset in patients after MDT (34% vs 8%, p = 0.06). The most recent ORIOLE phase 2 trial investigated the progression rate at 6 months after SBRT for up to 3 metastases [24]. Although PSMA-PET/CT was performed at baseline, it was blinded to the radiation oncologist so that in some patients not all PSMA-avid lesions were treated. The intervention arm showed a significantly reduced progression rate of 19% vs 61% (p = 0.005). Furthermore, patients with no additional PSMA-avid lesions at baseline had longer distant metastasis free survival (29 months vs 6 months,

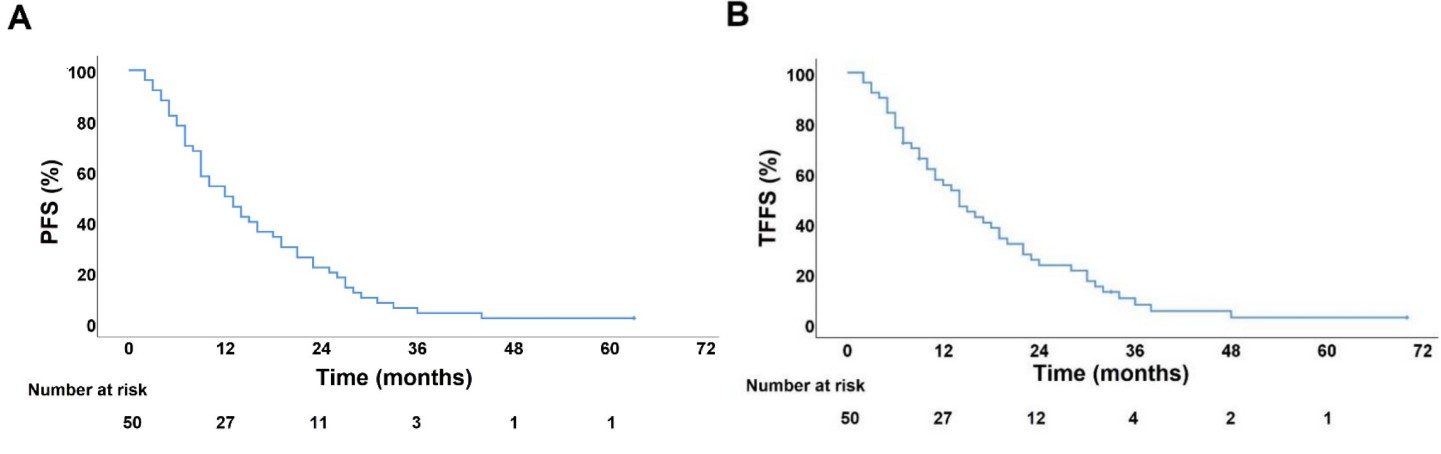

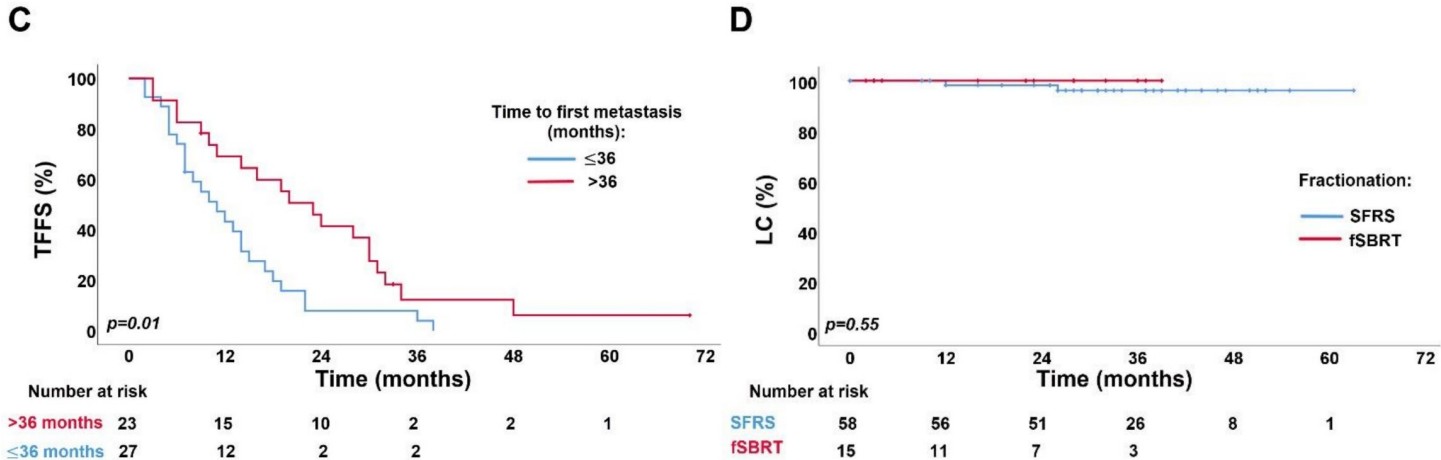

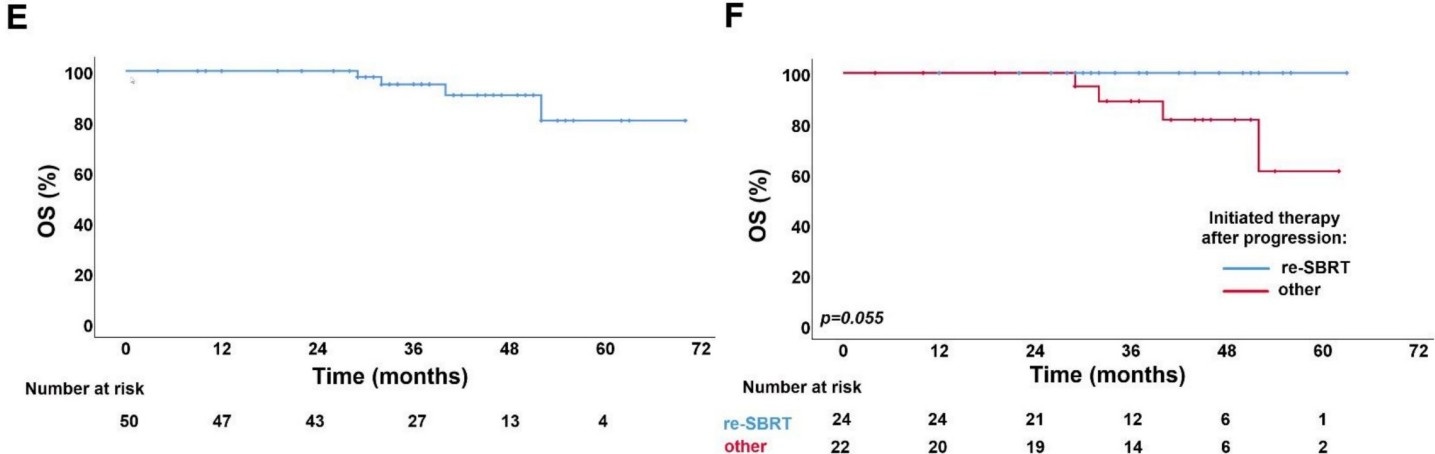

**Fig 2.** Kaplan–Meier survival curves for: (A) progression-free survival (PFS), (B) treatment failure-free survival (TFFS) (C) treatment failure-free survival by time from PCA diagnosis to first metastasis: >36 months vs ≤36 months, (D) local control (LC) by fractionation schedules: single fraction radiosurgery (SFRS) vs fractionated stereotactic body radiotherapy (fSBRT), (E) overall survival (OS), (F) overall survival by therapy initiated after progression: repeated SBRT (re-SBRT) vs other.

**Table 2. Progression pattern and therapy initiated in case of treatment failure in all patients.**

| Progression pattern | Number (%) | Therapy in case of TF | Number (%) |
|---|---|---|---|
| **Repeat OMPC (5 ≤ metastases)** | 32 (64) | | |
| | | SBRT | 22 (68.8) |
| | | ADT initiation | 7 (21.9) |
| | | ADT escalation | 1 (3.1) |
| | | combined | 1 (3.1) |
| | | no | 1 (3.1) |
| **Polymetastatic disease (5 > metastases)** | 6 (12) | | |
| | | ADT initiation | 4 (66.7) |
| | | ADT escalation | 2 (33.3) |
| **Biochemical (PSA) progression** | 6 (12) | | |
| | | ADT initiation | 3 (50) |
| | | ADT escalation | 3 (50) |
| **In-field progression** | 2 (4) | | |
| | | SBRT | 1 (50) |
| | | Surgery | 1 (50) |
| **Prostate/prostatic lodge recurrence** | 3 (6) | | |
| | | SBRT | 1 (33.3) |
| | | no | 2 (66.7) |
| **No progression** | 1 (2) | no | 1 (100) |

Abbreviations: ADT = androgen deprivation therapy; OMPC = oligometastatic prostate cancer; PSA = prostate-specific antigen; SBRT = stereotactic body radiotherapy; TF = treatment failure.

p = 0.0008), suggesting that PSMA-PET/CT-based SBRT may not only serve to treat existing metastases, but may also modulate course of disease.

In our analysis, the majority of patients (64%) with a progression after SBRT developed up to five new metastases and were therefore still considered to have a repeat OMPC. Other authors reported similar results, with 70–75% of patients treated with SBRT remaining oligo-progressive or oligorecurrent after distant relapse with median ≤3 metastases [25, 26]. This implies that in case of progression most patients are still eligible for further MDT.

Median PFS reported in the literature varies from 3 to 24 months (Table 4). Some authors observed a 21-month difference in median PFS in castration-sensitive versus castration-resistant patients with a maximum 3 bone metastases [20]. Furthermore, another small series found 1-year PFS rates to be 67% vs 0% in castration-sensitive compared to castration-resistant patients after radiotherapy to a maximum 3 metastases [18]. Such a difference in PFS between the groups raises the question of whether patients with progression despite hormone therapy are suitable candidates for MDT alone. However, in some castration-resistant patients, progression is limited only to a few sites, while the remaining disease is controlled by systemic therapy. In this case, the eradication of castration-resistant metastases using MDT allows a continuation of ongoing ADT and thus spares a second-line hormone or chemotherapy for further progression [27]. In contrast to the aforementioned studies, Valeriani and colleagues observed a relatively high median PFS of 18.4 months in 29 castration-resistant patients with oligoprogressive PCA treated with local radiotherapy for up to 3 metastases [28]. In present study, no differences in PFS in patients with or without ADT at the time of SBRT was observed.

In the case of repeat OMPC, multiple SBRT might be used as a bridging treatment to delay palliative system therapy. Recently, prospective analysis of 199 OMPC patients (76.4% staged

**Table 3. Univariate and multivariate analysis of factors influencing PFS and TFFS.**

| | PFS | | TFFS | | | |
| | Univariable | | Univariable | | Multivariable | |
| Determinant | HR (95% CI) | p-value | HR (95% CI) | p-value | HR (95% CI) | p-value |
|---|---|---|---|---|---|---|
| **Time from PCA to first metastasis (months)** | | | | | | |
| > 36 | 1 | | 1 | | 1 | |
| ≤ 36 | 2.17 (1.20–3.91) | 0.01 | 2.18 (1.18–4.02) | 0.01 | 2.54 (1.33–4.82) | 0.01 |
| **Gleason score** | | | | | | |
| ≤7 | 1 | | 1 | | N.A. | |
| >7 | 1.20 (0.66–2.17) | 0.55 | 0.99 (0.73–1.33) | 0.96 | | |
| **Primary tumor size** | | | | | | |
| T ≤ 2 | 1 | | 1 | | N.A. | |
| T > 2 | 0.98 (0.53–1.79) | 0.95 | 0.99 (0.53–1.84) | 0.99 | | |
| **Regional lymph node involvement at PCA diagnosis** | | | | | | |
| N0 | 1 | | 1 | | N.A. | |
| N1 | 1.50 (0.75–3.00) | 0.25 | 1.66 (0.83–3.32) | 0.15 | | |
| **Initial PSA (ng/ml)** | | | | | | |
| ≤ 10 | 1 | | 1 | | N.A. | |
| > 10 | 0.89 (0.47–1.60) | 0.65 | 0.91 (0.49–1.67) | 0.75 | | |
| **PSA (ng/ml) before SBRT** | | | | | | |
| ≤ 1 | 1 | | 1 | | 1 | |
| > 1 | 1.69 (0.88–3.22) | 0.11 | 1.02 (0.99–1.05) | 0.06 | 2.25 (1.10–4.59) | 0.03 |
| **Salvage radiotherapy after prostatectomy** | | | | | | |
| Yes | 1 | | 1 | | N.A. | |
| No | 1.27 (0.71–2.27) | 0.43 | 1.27 (0.70–2.31) | 0.43 | | |
| **Concomitant ADT** | | | | | | |
| Yes | 1 | | 1 | | N.A. | |
| No | 1.57 (0.84–2.93) | 0.16 | 1.69 (0.88–3.25) | 0.11 | | |
| **Number of metastases at SBRT** | | | | | | |
| 1 | 1 | | 1 | | 1 | |
| >1 | 1.54 (0.84–2.83) | 0.16 | 1.70 (0.91–3.17) | 0.10 | 1.42 (0.73–2.73) | 0.30 |
| **Number of affected organs** | | | | | | |
| 1 | 1 | | 1 | | N.A. | |
| >1 | 1.53 (0.54–4.34) | 0.42 | 1.72 (0.61–4.90) | 0.31 | | |
| **Bone metastases** | | | | | | |
| No | 1 | | 1 | | N.A. | |
| Yes | 0.81 (0.46–1.42) | 0.46 | 0.84 (0.47–1.51) | 0.56 | | |
| **Extra-pelvic lymph node metastases** | | | | | | |
| No | 1 | | 1 | | N.A. | |
| Yes | 0.75 (0.32–1.73) | 0.50 | 0.62 (0.27–1.48) | 0.29 | | |

Abbreviations: CI = confidence interval; HR = hazard ratio; NA = not assessed; PCA = prostate cancer; PFS = progression free-survival; PSA = prostate-specific antigen; SBRT = stereotactic body radiotherapy; TFFS = treatment failure-free survival.

with PSMA-PET/CT) with ≤5 metastases after SBRT reported 31.7%, 9.5% and 4% of patients receiving second, third, and fourth courses of SBRT [29]. After a median follow-up of 35.1 months, the majority of patients (51.7%) did not require a further tumor directed therapy. In 49.3% of patients palliative systemic- or radiotherapy had been postponed for a median time of 27.1 months (95% CI 21.8–29.4). Bouman–Wammes *et al.* investigated the impact of SBRT

**Table 4. Studies on SBRT for OMPC patients.**

| Reference | Year | No. of patients/ met. | No. of met. | Met. location | Castration sensitivity | Radiotherapy | Treatment outcomes | |
|---|---|---|---|---|---|---|---|---|
| | | | | | | | PFS | ADTFS |
| Prospective | | | | | | | | |
| Phillips et al. (ORIOLE) [24] | 2020 | 54/72 | ≤3 | LN = 33% | 100% | SBRT with 19.5 to 48.0 Gy in 1 to 3 fractions | Median in SBRT arm was not reached after 18.8 months of FU vs 5.8 months in observation arm | N.A. |
| | | | | Bone = 21% | | | | |
| Siva et al. [21] | 2018 | 33/50 | ≤3 | LN = 36.4% | 67% | SFRS with 20 Gy | 1-yr: 58% | 2-yr: 48% |
| | | | | Bone = 60.6% | | | 2-yr: 39% | |
| | | | | Both = 3.0% | | | | |
| Ost et al. (STOMP) [2] | 2017 | 62/116 | ≤3 | LN = 54.8% | 100% | SBRT in 80.6% | Median 10 months in MDT arm vs 6 months in surveillance arm | Median 21 months in MDT arm vs 13 months in surveillance arm |
| | | | | Non-nodal = 45.2% | | | | |
| Retrospective | | | | | | | | |
| Hurmuz et al. [31] | 2020 | 176/353 | ≤5 | LN = 34.7% | Unknown | SBRT in 73% with median 27 Gy in median 3 fractions; Conventional RT in 27% with median 60 Gy | Median 39.3 months | N.A. |
| | | | | Bone = 42.6% | | | | |
| | | | | Both = 22.7% | | | 2-yr: 63.1%, | |
| Nicosia et al. [32] | 2020 | 109/155 | ≤5 | LN = 100% | 100% | SBRT with median 36 Gy in 4–7 fractions | Median 14.5 months | Median 15 months |
| | | | | | | | 1-yr: 54.6% | |
| | | | | | | | 2-yr: 32.8%, | |
| Oehus et al. [33] | 2020 | 78/185 | ≤5 | LN = 68.2% | Unknown | SBRT in 20.5% | Median: 17.0 months | Median not reached after 16 months of follow-up |
| | | | | Bone = 45% | | | | |
| | | | | Visceral = 6.5% | | | 1-yr: 55.3%, | |
| Franzese et al. [34] | 2019 | 92/119 | ≤5 | LN, bone and visceral | 66% | SBRT with median 42 Gy in 2 to 8 fractions | Median 9.4 months | N.A. |
| | | | | | | | 1-yr: 42.8% | |
| | | | | | | | 3-yr: 16.7%, | |
| Patel et al. [20] | 2019 | 51/64 | ≤3 | Bone = 100% | 82% | SBRT with 24 to 30 Gy in 3 or 5 fractions | Median 24 months in castration sensitive vs 3 months in castration resistant | N.A. |
| Valeriani et al. [28] | 2019 | 29/37 | ≤3 | LN = 5.4% | 0% | SBRT for 16.2% | Median 18,4 months | N.A. |
| | | | | Bones = 83.8% | | | | |
| | | | | Other = 10.8% | | | 2-yr: 38.3% | |
| | | | | | | | 3-yr: 8.5%, | |
| Ong et al. [19] | 2019 | 20/26 | ≤3 | LN = 75% | 100% | SBRT with 30 Gy in 3 fractions and 35 to 40 Gy in 5 fractions | 1-yr: 62% | 1-yr: 70% |
| | | | | Bone = 15% | | | | |
| | | | | Both = 10% | | | | |
| Guler et al. [18] | 2018 | 23/38 | ≤3 | LN = 44.7% | 57% | Hypofractionated RT | 1-yr: 51% | N.A. |
| | | | | Bone = 55.3% | | | | |
| Triggiani et al. [35] | 2017 | 141/209 | ≤3 | LN = 79% | 71% | SBRT with 24 to 45 Gy in 3 to 6 fractions | Median in castration sensitive 17.7 months vs 11 months in castration resistant | Median ADTFS 20.9 months in castration sensitive vs median ADTEFS 22 months in castration resistant |
| | | | | Bone = 21% | | | | |
| Bouman-Wammes et al. [30] | 2017 | 43/54 | ≤4 | LN = 76.6% | 100% | SBRT with 30 or 35 Gy in 3 or 5 fractions | N.A. | Median 15.6 months |
| | | | | Bone = 20.9% | | | | |
| | | | | Both = 2.3% | | | | |
| Pasqualetti et al. [36] | 2016 | 29/45 | ≤3 | LN = 55.5% | 62% | SBRT with 24 Gy or 27 Gy in 1 or 3 fractions | N.A. | Median (systemic therapy free survival) 39.7 months |
| | | | | Bone = 44.5% | | | | |

(*Continued*)

**Table 4.** (Continued)

| Reference | Year | No. of patients/ met. | No. of met. | Met. location | Castration sensitivity | Radiotherapy | Treatment outcomes | |
|---|---|---|---|---|---|---|---|---|
| | | | | | | | PFS | ADTFS |
| **Decaestecker et al. [25]** | 2014 | 50/70 | ≤3 | LN = 54% | 100% | SBRT with 30 or 50 Gy in 3 or 10 fractions | Median 19 months | Median 25 months |
| | | | | Bone = 44% | | | | 1-yr: 82% |
| | | | | Visceral = 2% | | | 1-yr: 64% | 2-yr: 60% |
| | | | | | | | 2-yr: 35%, | |
| **Current study** | 2020 | 50/75 | ≤5 | LN = 48% | 70% | SFRS 80% with median 20 Gy | Median 12 months | Median not reached |
| | | | | Bone = 46% | | | | |
| | | | | Both = 4% | | | 1-yr: 54% | 1-yr: 76% |
| | | | | Visceral = 2% | | | 2-yr: 22% | 2-yr: 60% |

Abbreviations: ADTFS = androgen deprivation therapy-free survival; LN = lymph node; MDT = metastasis directed therapy; N.A. = not assessed;
OMPC = oligometastatic prostate cancer; RT = radiotherapy; SBRT = stereotactic body radiotherapy; SFRS = single fraction radiosurgery.

on delaying ADT for 43 hormone-sensitive PCA patients with <5 metastases detected using choline-PET-CT [30]. The second SBRT course was applied in 16.3% of patients with a median 19.8 months between the courses, which is in line with our results. The median ADTFS observed within this group was 32.1 months (95% CI: 7.8–56.5). Furthermore, Triggiani and colleagues observed a 18% rate of repeated SBRT in 141 patients with hormone-sensitive and castration-resistant OMPC treated with SBRT for up to 3 metastases [35]. In our cohort, a second SBRT course was the treatment of choice in almost 50% of patients with progression and thus ADT initiation or escalation was delayed. The median ADTFS was not reached after 34 months follow-up. Furthermore, we observed a trend (p = 0.055) toward better OS after second SBRT course compared to other therapy initiated after progression.

The median ADTFS reported in the literature for patients with OMPC after MDT varies between 20.9 and 39.7 months, which is comparable to our results Table 4. However, the results of different studies should be compared with caution, due to diverse inclusion criteria (e.g. number of metastases), staging methods (PSMA/PET-CT, FDG/PET-CT), treatment modalities (SBRT, surgery) and different indications for ADT start used.

In our analysis SFRS showed excellent LC rates of 96% at 2 years with no grade ≥3 adverse events. Siva *et al.* prospectively analyzed safety and feasibility of SFRS with 20 Gy for bone and lymph node metastases staged with sodium fluoride PET/CT. After treating 50 lesions in 33 patients, the authors observed 1- and 2-year LC rates of 97% and 93%, respectively. Grade 3 adverse events were observed in one patient (3%) [21]. Muldermans *et al.* reported LC at 2 years of 82% after treating 69 patients with 81 metastases– 88% received SFRS with a median dose of 16 Gy (range: 16–24) [37]. Seventy percent of patients were staged with choline PET/ CT. In multivariate analysis, radiation dose ≥18 Gy was associated with better LC. No grade ≥2 adverse events were observed. Although, the prescribed dose varied within the studies emerging data including our study show that SFRS can be safely used in favor of patients' convenience and provide excellent LC rates.

In our analysis a TTM of more than 36 months was found to be an independent prognostic factor for prolonged TFFS and was associated with greater PFS. Benefits might be explained by an indolent tumor biology with a lower metastatic potential. Supporting this hypothesis, analysis of a multi-institutional study on oligometastatic disease from several tumor entities showed that longer TTM using the MDT-approach resulted in improved survival [38].

The retrospective study design, relatively small sample size including heterogeneous patients, inherent patient selection bias and lack of control group are the major limitations of our study. Furthermore, the comparison between SFRS and fSBRT group needs to be interpreted with caution due to limited number of metastases treated with fSBRT. The majority of patients had a high risk PCA, so conclusions for patients with low and medium risk of PCA should be drawn carefully. Nonetheless, we were able to show the efficacy, safety, and excellent local control rates after SFRS use in OMPC patients.

## Conclusions

In conclusion, our study suggests that PSMA-PET/CT-based SFRS might be considered a valid treatment option for OMPC patients, including cases with repeat oligometastatic disease. This way, the onset or escalation of palliative ADT and its potential side effects can be avoided. Metastases treated with SFRS reached excellent local control rates with minimal toxicity. Low PSA levels and longer TTM predicts elongated TFFS. Randomized studies are needed to support our findings.

## Supporting information

**S1 Table. Metastases location.**
(DOCX)

**S2 Table. Treatment characteristic.**
(DOCX)

## Author Contributions

**Conceptualization:** Goda Kalinauskaite, Markus Kufeld, Ingeborg Tinhofer, Volker Budach, Arne Grün, Carmen Stromberger.

**Data curation:** Goda Kalinauskaite, Carolin Senger, Markus Kufeld, Arne Grün, Carmen Stromberger.

**Formal analysis:** Goda Kalinauskaite, Arne Grün, Carmen Stromberger.

**Funding acquisition:** Goda Kalinauskaite.

**Investigation:** Goda Kalinauskaite, Carmen Stromberger.

**Methodology:** Goda Kalinauskaite, Ingeborg Tinhofer, Volker Budach, Arne Grün, Carmen Stromberger.

**Project administration:** Goda Kalinauskaite, Arne Grün.

**Resources:** Goda Kalinauskaite, Carolin Senger, Marcus Beck, Carmen Stromberger.

**Software:** Goda Kalinauskaite, Anne Kluge.

**Supervision:** Goda Kalinauskaite, Carolin Senger, Arne Grün, Carmen Stromberger.

**Validation:** Goda Kalinauskaite, Alexandra Hochreiter.

**Visualization:** Goda Kalinauskaite, Anne Kluge.

**Writing – original draft:** Goda Kalinauskaite.

**Writing – review & editing:** Goda Kalinauskaite, Carolin Senger, Anne Kluge, Christian Furth, Markus Kufeld, Ingeborg Tinhofer, Volker Budach, Marcus Beck, Alexandra Hochreiter, Arne Grün, Carmen Stromberger.

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
