## [Decision Letter · Decision Letter 0]

14 Jul 2020

PONE-D-20-17881

68Ga-PSMA-PET/CT-based radiosurgery and stereotactic body radiotherapy for oligometastatic prostate cancer

PLOS ONE

Dear Dr. Kalinauskaite,

Thank you for submitting your manuscript to PLOS ONE. After careful consideration, we feel that it has merit but does not fully meet PLOS ONE’s publication criteria as it currently stands. Therefore, we invite you to submit a revised version of the manuscript that addresses the points raised during the review process.

We look forward to receiving your revised manuscript.

Kind regards,

Stephen Chun

Academic Editor

PLOS ONE

Journal Requirements:

Reviewers' comments:

Reviewer's Responses to Questions

**Comments to the Author**

1. Is the manuscript technically sound, and do the data support the conclusions?

Reviewer #1: Yes

Reviewer #2: Yes

2. Has the statistical analysis been performed appropriately and rigorously? 

Reviewer #1: Yes

Reviewer #2: Yes

3. Have the authors made all data underlying the findings in their manuscript fully available?

Reviewer #1: Yes

Reviewer #2: No

4. Is the manuscript presented in an intelligible fashion and written in standard English?

Reviewer #1: Yes

Reviewer #2: Yes

5. Review Comments to the Author

Reviewer #1: The article presented by the authors describes outcomes following SRS or FSBRT for oligorecurrent prostate cancer at a single institution. This article adds to the growing body of literature supporting the use of metastases directed therapy as a viable treatment option to delay initiation (or escalation of dose) of androgen deprivation therapy. What makes the data in this manuscript unique relative to the other publications is that the metastatic lesions were diagnosed and treated using PSMA-PET.

My critiques and recommendations are as follows:

1) The term oligometastatic should be replaced with “oligorecurrent” in the manuscript when describing the patient population, including the title. Please see recent publication in Lancet by ESMO about standardizing nomenclature for oligometastatic biology and try to be consistent with this terminology in the introduction/discussion as well. For example, in line 205 of the discussion, “patients treated with SBRT remaining oligometastatic” is not correct and should be changed to “oligoprogressive.”

Characterisation and Classification of Oligometastatic Disease: A European Society for Radiotherapy and Oncology and European Organisation for Research and Treatment of Cancer Consensus Recommendation. Lancet Oncol . 2020 Jan;21(1):e18-e28. doi: 10.1016/S1470-2045(19)30718-1.

2) In line with the above, in the study population section of Materials and Methods the authors shoould clearly state that the data is of patients treated with curative local therapy, who then relapsed/recurred with a median time to recurrence of x months. The way it is currently written implies that they may or may not have had metastases at diagnosis. These data are important to inform the reader, especially since time from PCA to first metastasis is a predictor that influences PFS and TFFS.

3) Under results, line 129, the percentage of “2 patients” is incorrectly listed as 2%, it should be 4%.

4) In materials and methods, the authors should state some reasons why fractionated SBRT was favored over single fraction SRS in some of the cases. Given this is a retrospective study and a comparison of local recurrence was made between the two treatments, the reader should be informed whether there may be any potential confounders that led the treating physicians to picking one vs the other. Were there any cases of reirradiation?

5) Why was overall survival not assessed as another potential endpoint? It would be nice to see this data along with PFS and TFFS, and any relevant univariate and multivariate analyses.

6) How did survival compare among patients treated with a second cousre of SBRT (w/ intent to defer ADT) vs those who had a dose escalation or initiation of ADT? This would be a helpful survival curve to see and can be analyzed relative to the two trials that are referenced in the discussion.

7) In the final paragraph, the authors should state other potential biases present in their retrospective study and additional considerations, other than the mentioned small sample size which they acknowledge.

Reviewer #2: The authors did a great job compiling this list of PSMA directed SBRT for OMPC. Please address the following:

- Please clarify whether these patients were hormone sensitive vs castration resistant. In other words, the 30% who were on ADT, are they castration resistant or did they start ADT and SBRT at same time?

- Table 1 shows that the minimal time to metastasis was 1 month. Does this mean that all patients are recurrent after primary treatment and that there is no patient with de novo OMPC? Did the authors intentionally excluded patients with limited mets at original presentation?

- What is the relevance of using TFFS as an endpoint vs PFS?

- How was in-field progression defined? Based on imaging? Were patients getting routine imaging for all irradiated lesions?

- There is a discrepancy between text and table 1: text says they treated 75 lesions whereas table one indicated 84? Please clarify.

- There is a discrepancy in the text and Table 1 regarding the fractionation schedules: table says 13 patients got 3 fraction SBRT and 2 patients got “other” while text says 10 and 5 patients respectively. Please clarify.

- Can you please specify what are the “other fractionations” in table 1?

- Could you please show numbers at risk for Kaplan Meier Curves shown in Figure 1

- Abstract says that 31 patients had further oligoprogression while Table 2 says it is 32. Please clarify.

- Would you please add a table summarizing the current available data on using SBRT in OMPC (hormone sensitive or castration resistant).

- Please reread the manuscript and make sure the numbers agree between text and tables

6. PLOS authors have the option to publish the peer review history of their article (what does this mean?). If published, this will include your full peer review and any attached files.

Reviewer #1: No

Reviewer #2: No

---

## [Author Response · Author response to Decision Letter 0]

21 Sep 2020

Dear Reviewers,

We thank you for the close and thoughtful reading of our manuscript, for the interest in our work, and for valuable suggestions for improvement. We revised our manuscript based on the constructive suggestions from you. We believe these revisions extend the significance of our conclusions. We provide point by point answers to your comments.

Sincerely Yours,

Goda Kalinauskaite

Department of Radiation oncology, 

Charité – Universitätsmedizin Berlin 

Phone: +49 15 234 778300

Email: goda.kalinauskaite@charite.de

Point by point reply:

Reviewer(s)' Comments to Author:

Reviewer #1: 

The article presented by the authors describes outcomes following SRS or FSBRT for oligorecurrent prostate cancer at a single institution. This article adds to the growing body of literature supporting the use of metastases directed therapy as a viable treatment option to delay initiation (or escalation of dose) of androgen deprivation therapy. What makes the data in this manuscript unique relative to the other publications is that the metastatic lesions were diagnosed and treated using PSMA-PET.

Point #1: The term oligometastatic should be replaced with “oligorecurrent” in the manuscript when describing the patient population, including the title. Please see recent publication in Lancet by ESMO about standardizing nomenclature for oligometastatic biology and try to be consistent with this terminology in the introduction/discussion as well. For example, in line 205 of the discussion, “patients treated with SBRT remaining oligometastatic” is not correct and should be changed to “oligoprogressive.” Characterisation and Classification of Oligometastatic Disease: A European Society for Radiotherapy and Oncology and European Organisation for Research and Treatment of Cancer Consensus Recommendation. Lancet Oncol . 2020 Jan;21(1):e18-e28. doi: 10.1016/S1470-2045(19)30718-1.

Reply #1: We are very grateful for this point and reference suggestion which revealed that some terms referring to oligometastatic disease used in our paper were not sufficient or incorrect. After careful study of recently published ESMO and ESTRO recommendation consensus regarding characterization and classification of oligometastatic disease we believe that umbrella term “oligometastatic prostate cancer’’ used in our paper in general describes the population of patients with limited number of metastases (≤5) and is in agreement with consensus: “After discussion, 17 (94%) of 18 participants agreed (either strongly agreed or agreed) on oligometastatic disease as the umbrella term for all states of limited metastatic disease, staying within the tradition of the original publication of Hellman and Weichselbaum.’’. We could not replace oligometastatic with oligorecurrent as suggested because our study population included all de-novo oligometastatic patients: synchronous oligometastatic (time between PCA diagnosis and first metastatsis < 6 months), metachronous oligorecurrence (time between PCA diagnosis and first metastatsis > 6 months without systemic treatment at the time of metastasis diagnosis) and metachronous oligoprogression (time between PCA diagnosis and first metastatsis > 6 months under systemic treatment at the time of metastasis diagnosis). In order to characterize the patients’ group in detail we rewrote the materials and methods, study population section, lines 82-88.

Further changes:

• Abstract, results, line 41: further oligoprogression changed into repeat oligometastatic disease;

• Results, line 155: distant oligoprogression changed into repeat oligometastatic disease;

• Table 2, 2. row: Oligometastatic disease changed into repeat OMPC;

• Discussion, line 208: considered to have an OMPC changed into a repeat OMPC

• Discussion, line 209-210: oligometastatic changed into oligoprogressive or oligorecurent;

• Discussion, line 223: oligorecurrent changed into oligoprogressive

• Conclusions, lines 273-274: further oligoprogression changed into repeat oligometastatic disease.

Point #2: In line with the above, in the study population section of Materials and Methods the authors should clearly state that the data is of patients treated with curative local therapy, who then relapsed/recurred with a median time to recurrence of x months. The way it is currently written implies that they may or may not have had metastases at diagnosis. These data are important to inform the reader, especially since time from PCA to first metastasis is a predictor that influences PFS and TFFS. 

Reply #2: To clarify our study population we changed the Materials and methods, Study population section, line 82-88 as suggested.

Point #3: Under results, line 129, the percentage of “2 patients” is incorrectly listed as 2%, it should be 4%.

Reply #3: We are very sorry for this mistake and assure you that we doublechecked all the numbers in the abstract, main text and tables. The mismatch of numbers was changed as suggested by the reviewer. Results section, line 143.

Point #4: In materials and methods, the authors should state some reasons why fractionated SBRT was favored over single fraction SRS in some of the cases. Given this is a retrospective study and a comparison of local recurrence was made between the two treatments, the reader should be informed whether there may be any potential confounders that led the treating physicians to picking one vs the other. Were there any cases of reirradiation? 

Reply #4: We are grateful for reviewer’s important point. We stated as suggested why we chose fSBRT over SFRS in Material and methods, Radiotherapy section, lines 107-110.

In our research we have excluded the patients who had had previous SBRT, no case with re-RT was included. See also method section: opulation characteristic section, line 87-88.

Point #5: Why was overall survival not assessed as another potential endpoint? It would be nice to see this data along with PFS and TFFS, and any relevant univariate and multivariate analyses.

Reply #5: Initially we excluded overall survival (OS) as a potential endpoint because most of the patients with PCA live long and the effect of therapy might be observed only after very long follow-up. In our population we observed 1- , 3- and 5-years OS rates of 100% and 94.7% and 80.3%. Only 4 patients were dead at the time of analysis. In uni- and multivariate cox analysis we did not observe any factors influencing the OS. However, we completely agree that this is an interesting endpoint and included it among secondary endpoints as suggested. 

Point #6: How did survival compare among patients treated with a second course of SBRT (w/ intent to defer ADT) vs those who had a dose escalation or initiation of ADT? This would be a helpful survival curve to see and can be analyzed relative to the two trials that are referenced in the discussion.

Reply #6: Thank you very much for very interesting suggestion. We divided 46 patients who received any new therapy after progression in 2 groups: group 1 patients treated with SBRT (n=24) and group 2 patients treated with other therapy (n=22). Four patients were excluded from the analysis because they either were free from progression, or the data on the therapy was lacking. All 4 patients who died are in group 2. Kaplan Meier analysis showed a trend (p=0.055) toward better survival in group treated with SBRT. We included these results (please find Kaplan Meier curve in attached response to reviewers letter) in our manuscript, Fig 2: F. We also compared patients who after progression had polymetastatic diseases with those who remained oligometastatic. We found that all patients with polymetastatic disease were alive at last follow-up and there is no survival difference between the groups (please find Kaplan Meier curve in attached response to reviewers letter). We understand that these results should be interpreted with caution due to the small sample size and the small number of events. 

We also compared patients who after progression had polymetastatic diseases with those who remained oligometastatic. We found that all patients with polymetastatic disease were alive at last follow-up and there is no survival difference between the groups. We understand that these results should be interpreted with caution due to the small sample size and the small number of events. 

Point #7: In the final paragraph, the authors should state other potential biases present in their retrospective study and additional considerations, other than the mentioned small sample size which they acknowledge.

Reply #7: Thank you for this point. As you suggested we discussed the further limitations such as patient selection bias, lack of control group and dominance of high-risk PCA. Discussion section, paragraph 9, Line 264-267.

Reviewer #2: 

The authors did a great job compiling this list of PSMA directed SBRT for OMPC. Please address the following:

Point #1:

Please clarify whether these patients were hormone sensitive vs castration resistant. In other words, the 30% who were on ADT, are they castration resistant or did they start ADT and SBRT at same time?

Reply #1: Thank you very much for this important question. All 15 patients with simultaneous ADT at the first SBRT were castration resistant. To evaluate the “true’’ effect of SBRT on irradiated lesions, we excluded those patients who started or changed ADT simultaneously with first SBRT. We are sorry for not mentioning this in the text, since it is an important aspect. We included this in materials and methods section, line 86-88. 

Point #2: Table 1 shows that the minimal time to metastasis was 1 month. Does this mean that all patients are recurrent after primary treatment and that there is no patient with de novo OMPC? Did the authors intentionally excluded patients with limited mets at original presentation?

Reply #2: All patients with either synchronous or metachronous metastases were included, however those who started ADT simultaneously with SBRT were excluded and this resulted into minimal time to metastases of one month. By excluding patients with simultaneous onset of ADT and SBRT, we were able to assess the "true" effect of SBRT.

Point #3: What is the relevance of using TFFS as an endpoint vs PFS?

Reply #3: TFFS was defined as: new tumor-directed therapy (e.g. repeated SBRT, start of ADT, escalation of an ongoing ADT, surgery, chemotherapy). Since many of the oligometastatic PCA patients often undergo further therapy, we wanted to investigate the new therapy-free interval after SBRT to reflect how much of the treatment-free time patients receive after SBRT. We find this point is important for patients quality of life. Especially in those patients who received multiple courses of therapy. However, if you think, that it does not add any value to the manuscript, we are willing to remove this endpoint.

Point #4: How was in-field progression defined? Based on imaging? Were patients getting routine imaging for all irradiated lesions?

Reply #4: In-field progression was defined as an increase of metastasis volume or local regrowth within any volume of the PTV. LC was assessed using conventional (CT or MRT) or functional (PSMA-PET/CT) imaging. To make it clear we included this in Materials and methods, endpoints section, line 125-126. The imaging was conducted not routinely but rather in case of PSA elevation. Your valuable comment pointed directly into inaccuracy we accidentally overlooked. No imaging was performed in two patients with elevated PSA, so that local control was available for 73 metastases. We have included this in the text in the results section, line 163.

Point #5: There is a discrepancy between text and table 1: text says they treated 75 lesions whereas table one indicated 84? Please clarify.

Reply #5: We are very sorry for this miscalculation. The number true number of metastases is 75 as listed in abstract and in the text. We changed the numbers in the table. We assure you that we double checked all the numbers in the abstract, text and the tables. 

Point #6: There is a discrepancy in the text and Table 1 regarding the fractionation schedules: table says 13 patients got 3 fraction SBRT and 2 patients got “other” while text says 10 and 5 patients respectively. Please clarify.

Reply #6: We are very sorry for this discrepancy. The numbers given in the table are correct. Two metastases received SBRT another fractionation schedule: 6 x 4.8 Gy and 5 x 5 Gy. 

Point #7: Can you please specify what are the “other fractionations” in table 1? To clarify treatment and metastases parameters we included 2 tables under supporting information section. 

Reply #7: Other fractionation schedules as noted above were: 6 x 4.8 Gy and 5 x 5 Gy.

Point #8: Could you please show numbers at risk for Kaplan Meier Curves shown in Figure 1.

Reply #8: We included numbers at risk in Figure 2 as requested. 

Point #9: Abstract says that 31 patients had further oligoprogression while Table 2 says it is 32. Please clarify.

Reply #9: The correct number of patients with repeated oligometastatic disease is 32 as stated in the text and Table 1. We correct this discrepancy and thank the reviewer for real careful revision of our manuscript. 

Point #10: Would you please add a table summarizing the current available data on using SBRT in OMPC (hormone sensitive or castration resistant).

Reply #10: We summarized the current available data on exciting studies investigating SBRT for oligometastases in hormone sensitive and castration resistant patients. We included 15 studies: 3 prospective trials and 12 retrospective that reported PFS or/and ADTFS as an endpoint. We also believe that this summery will increase the value of our work. 

Point #11: Please reread the manuscript and make sure the numbers agree between text and tables.

Reply #11 We read the manuscript carefully and double-checked the numbers between text and the tables and figures.

---

## [Decision Letter · Decision Letter 1]

6 Oct 2020

68Ga-PSMA-PET/CT-based radiosurgery and stereotactic body radiotherapy for oligometastatic prostate cancer

PONE-D-20-17881R1

Dear Dr. Kalinauskaite,

We’re pleased to inform you that your manuscript has been judged scientifically suitable for publication and will be formally accepted for publication once it meets all outstanding technical requirements.

Kind regards,

Stephen Chun

Academic Editor

PLOS ONE

Additional Editor Comments (optional):

Reviewers' comments:

Reviewer's Responses to Questions

**Comments to the Author**

1. If the authors have adequately addressed your comments raised in a previous round of review and you feel that this manuscript is now acceptable for publication, you may indicate that here to bypass the “Comments to the Author” section, enter your conflict of interest statement in the “Confidential to Editor” section, and submit your "Accept" recommendation.

Reviewer #1: All comments have been addressed

Reviewer #2: All comments have been addressed

2. Is the manuscript technically sound, and do the data support the conclusions?

Reviewer #1: Yes

Reviewer #2: Yes

3. Has the statistical analysis been performed appropriately and rigorously? 

Reviewer #1: Yes

Reviewer #2: Yes

4. Have the authors made all data underlying the findings in their manuscript fully available?

Reviewer #1: Yes

Reviewer #2: No

5. Is the manuscript presented in an intelligible fashion and written in standard English?

Reviewer #1: Yes

Reviewer #2: No

6. Review Comments to the Author

Reviewer #1: (No Response)

Reviewer #2: Thank you so much for addressing the revision. Please review paper again for grammatical issues, style, and punctuation.

7. PLOS authors have the option to publish the peer review history of their article (what does this mean?). If published, this will include your full peer review and any attached files.

Reviewer #1: No

Reviewer #2: No

---

## [Editor Report · Acceptance letter]

12 Oct 2020

PONE-D-20-17881R1 

68Ga-PSMA-PET/CT-based radiosurgery and stereotactic body radiotherapy for oligometastatic prostate cancer 

Dear Dr. Kalinauskaite:

I'm pleased to inform you that your manuscript has been deemed suitable for publication in PLOS ONE. Congratulations! Your manuscript is now with our production department. 

Kind regards, 

on behalf of

Dr. Stephen Chun 

Academic Editor

PLOS ONE